# Investigation of the Seroprevalence of Brucella Antibodies and Characterization of Field Strains in Immunized Dairy Cows by *B. abortus* A19

**DOI:** 10.3390/vetsci11070288

**Published:** 2024-06-28

**Authors:** Yong Shi, Yimeng Cui, Gaowa Wudong, Shengnan Li, Ye Yuan, Danyu Zhao, Shurong Yin, Ziyang Diao, Bin Li, Dong Zhou, Xuejun Li, Zhanlin Wang, Fengxia Zhang, Min Xie, Zehui Zhao, Aihua Wang, Yaping Jin

**Affiliations:** 1College of Veterinary Medicine, Northwest A&F University, Yangling District, Xianyang 712100, China; hongxuelan@nwafu.edu.cn (Y.S.); cui_yimeng@nwafu.edu.cn (Y.C.); wudonggaowaa@163.com (G.W.); lisn9912@163.com (S.L.); 15637737955@163.com (Y.Y.); zhaodanyu@nwafu.edu.cn (D.Z.); yinshurong2442@163.com (S.Y.); 13884704285@163.com (Z.D.); zhoudong1949@163.com (D.Z.); 2Key Laboratory of Animal Biotechnology of the Ministry of Agriculture, Northwest A&F University, Yangling District, Xianyang 712100, China; 3Animal Health Supervision Institute of Lingwu City, Lingwu 750400, China; lwdwwsids_2006@126.com; 4Animal Disease Control and Prevention Center of Lingwu City, Lingwu 750400, China

**Keywords:** Brucella, Brucellosis, persistent antibodies, *Brucella abortus* strain A19

## Abstract

**Simple Summary:**

*Brucella abortus* strain 19 vaccines, including A19 and S19 vaccines, are widely used to prevent bovine brucellosis. Vaccine-induced antibodies are usually considered to last for no more than 12 months. After this period, individuals testing positive are excluded because they are presumed to be infected with Brucella field strains. In Ningxia, China, immunization with the *Brucella abortus* strain A19 vaccine is being implemented as part of a brucellosis eradication program. We conducted a serological survey of cattle immunized with the A19 vaccine for over 12 months in Lingwu, Ningxia, from 2021 to 2023. It was found that there was a certain proportion of cows with a persistent antibody titer who were not infected with brucellosis. Therefore, cattle with positive antibodies cannot be eliminated through simple serological detection. Otherwise, a large number of healthy cattle would be mistakenly culled, resulting in significant economic losses. We have also developed a PCR method to distinguish Brucella strain 19 from non-19 Brucella strains and successfully detected 10 *Brucella abortus* field strains from five dairy farms.

**Abstract:**

(1) Background: One method of eradicating brucellosis is to cull cattle that test positive for antibodies 12 months after being vaccinated with the 19-strain vaccine. Variations in immunization regimens and feeding practices may contribute to differences in the rate of persistent antibodies. We conducted this study to investigate the real positive rate of Brucella antibody in field strains of *Brucella* spp. after immunization over 12 months in dairy cows. This research aims to provide data to support the development of strategies for preventing, controlling, and eradicating brucellosis. (2) Method: We employed the baseline sampling method to collect samples from cows immunized with the A19 vaccine for over 12 months in Lingwu City from 2021 to 2023. Serological detection was conducted using the RBPT method. An established PCR method that could distinguish between 19 and non-19 strains of Brucella was utilized to investigate the field strains of Brucella on 10 dairy farms based on six samples mixed into one using the Mathematical Expectation strategy. (3) Results: We analyzed the rates of individual seropositivity and herd seropositive rates in dairy cattle in Lingwu City from 2021 to 2023 and revealed that antibodies induced by the *Brucella abortus* strain A19 vaccine persist in dairy herds for more than 12 months. We established a PCR method for identifying both Brucella A19 and non-A19 strains, resulting in the detection of 10 field strains of *Brucella abortus* from 1537 dairy cows. By employing a Mathematical Expectation strategy, we completed testing of 1537 samples after conducting only 306 tests, thereby reducing the workload by 80.1%. (4) Conclusions: There was a certain proportion of cows with a persistent antibody titer, but there was no evidence that all of these cattle were naturally infected with Brucella. The established PCR method for distinguishing between *Brucella abortus* strain 19 and non-19 strains can be specifically utilized for detecting natural Brucella infection in immunized cattle. We propose that relying solely on the detection of antibodies in cattle immunized with the A19 vaccine more than 12 months previously should not be solely relied upon as a diagnostic basis for brucellosis, and it is essential to complement this approach with PCR analysis to specifically identify field *Brucella* spp. *Brucella abortus* was the predominant strain identified in the field during this study. Detection based on the Mathematical Expectation strategy can significantly enhance detection efficiency.

## 1. Introduction

Brucellosis is caused by bacteria of the genus Brucella [1,2], which can lead to a potentially debilitating chronic infection associated with febrile illness in humans, abortion, premature birth, and decreased productivity in livestock [3,4]. Thus, brucellosis is a serious public health threat and is associated with significant economic losses in the livestock industry [5,6]. Over 500,000 new human cases of brucellosis are diagnosed each year [7,8], although the true number of cases is likely to be much higher due to inaccurate diagnosis, inadequate surveillance, and incomplete reporting [9]. Brucellosis is widespread throughout the world; only a few countries in the world have successfully achieved brucellosis eradication [10].

*Brucellae* are non-motile, non-spore-forming, and slow-growing Gram-negative coccobacilli belonging to the *Brucellaceae* family in the alpha-2 subclass of the proteobacteria [11,12]. In recent years, 12 species of Brucella have been identified [13]. Brucellosis is transmitted horizontally or vertically among both domestic and feral animals, and person-to-person transmission of the infection is rare [14]. *B. melitensis*, *B. abortus*, *B. suis*, and *B. canis* are the main causes of the disease in humans, with infections resulting from direct contact with birth fluids or infected tissues, or Brucella aerosol [15]. Consumption of unpasteurized dairy products infected with *Brucella* spp. is also a major route for human infection [16,17].

A common implication of herd immunity is that the risk of infection among susceptible individuals in a population is reduced by immunity accumulated naturally or through vaccination [18,19]. Control of zoonotic diseases in human populations has relied heavily on the control of animal disease [20]. Vaccination is the main strategy for the prevention and control of brucellosis [21,22]. *B. abortus* S19, RB51, and *B. melitensis* Rev. 1 vaccines have been widely used in many countries [23,24,25], while *B. abortus* A19, *B. melitensis* M5, and *B. suis* S2 vaccines are used in China [26,27,28].The strains A19 and S19 are predominantly utilized for Brucella vaccination in cattle [28]. Both originated from the *Brucella abortus* strain 19, isolated in the United States in 1923. Their genomes exhibit 99.9% homology, with S19 having a 702 bp deletion in the erythritol gene, which gene is complete in A19 [29]. Although Brucella live-attenuated vaccines can cause abortions in pregnant animals and are virulent in humans, this approach has played an important part in controlling the spread of epidemics and reducing the incidence of human disease [30]. However, most vaccines are made with Brucella smooth strains, as most serologic tests are primarily based on the detection of antibodies against the O-side chain of Brucella. Discriminating between vaccine-induced antibodies and those generated by newly generated field strains poses a challenge [30,31], making it necessary to use direct detection techniques such as bacterial culture and nucleic acid amplification for the diagnosis of brucellosis in animals.

The vaccination-test-slaughter eradication strategy is commonly employed for brucellosis control in many countries [10]. Typically, the presence of antibodies is detected for a period of time after vaccination, and antibody-positive livestock are subsequently culled. In China, according to the Technical Points for Brucellosis Prevention and Control (1st edition) issued by the Ministry of Agriculture of the People’s Republic of China, cattle should be monitored for A19 vaccine antibodies for a period of 12 months post-immunization. This prescribed approach implicitly recognizes the loss of antibodies 12 months post-immunization with the A19 vaccine. However, it has been reported that vaccine-induced antibodies persist in cattle immunized with the S19 vaccine 12 months after vaccination, making it a challenge to differentiate between vaccine-induced antibodies and those resulting from field strain infection [32,33]. Consequently, if cattle continue to test positive for antibodies beyond 12 months, indiscriminate culling may lead to the erroneous elimination of cattle that are not infected with *Brucella* spp., causing significant economic losses.

Similar to other provinces in northwest China where brucellosis is highly endemic, a livestock immunization program with the brucellosis vaccine has been implemented in NingXia Province. However, there is a paucity of reports regarding the seropositivity rate of anti-Brucella antibodies and the prevalence of field strains in livestock within the areas where this strategy has been implemented. The absence of this data severely hampers the development of effective prevention and control strategies for brucellosis in these regions, particularly human brucellosis [34]. The Chinese government has placed great emphasis on the prevention and control of brucellosis, as evidenced by the implementation of the Five-Year Action Plan for Brucellosis Prevention and Control in Livestock (2022–2026), which aims to achieve control of the individual positive rate at <0.4% and the herd positive rate at <7%. In 2022, the dairy cow population in Lingwu City reached 196,400 and produced an annual output of 567,900 tons of milk, ranking second in NingXia Province. The expansion of dairy farming in recent years has significantly heightened the risk of brucellosis transmission [35,36,37]. Epidemiological investigation is the first step to achieving the goal of brucellosis eradication. The objective of this study was to determine the seroprevalence of Brucella antibodies, characterize field *Brucella* spp. in immunized dairy herds over 12 months, and discuss the diagnostic criteria for brucellosis in cattle, thereby providing a scientific foundation for epidemiological investigation, control, and eradication of brucellosis in this region as well as other areas where immunization is implemented.

## 2. Materials and Methods

### 2.1. Study Design

The serology research project ran from March 2021 to July 2023. The serological survey included all large-scale farms and small farmers raising dairy cows in Lingwu City. In Lingwu City, cattle are typically first immunized with the Brucella A19 vaccine between 3 and 8 months of age, followed by a low-dose booster 3 months later via subcutaneous injection. Brucella vaccine A19 was produced by Qingdao Yibang Biological Co., Ltd. (Qingdao, China) or Xinjiang Tiankang Biological Co., Ltd. (Urumqi City, China). Cows aged between 1.5 and 6 years old who had been vaccinated with the Brucella A19 vaccine for over a year were selected. All the sampled cows were treated as independent individuals, while all the cows from the same dairy farm or dairy farming household were considered a herd. The sample plan was implemented following the baseline survey technique utilized in the Brucellosis Special Surveillance Program of the local government: Three samples were collected from each small dairy farming household, and at least twenty samples were collected from each large dairy farm. The individual positive rate and herd positive rate of Brucella antibodies were calculated during the three years. In addition, three brucellosis-free dairy farms were also tested to verify the antibody persistence of the A19 vaccine.

The Brucella pathogen survey was conducted from March to May 2023. A total of 10 pastures were randomly selected from 4 small farming areas (4 farms) and 3 large farming areas (6 farms) in Lingwu City. Then Brucella field strains were investigated on 10 dairy farms. The strains were clarified using the PCR method to distinguish Brucella A19 from other strains identified in this work.

### 2.2. Serological Tests

Whole blood (2–5 mL) was collected from the tail vein of the selected livestock. Samples were incubated at 37 °C for 1 h and then centrifuged at 3000 rpm for 5 min. The sera were decanted and stored at −20 °C prior to testing with the Rose Bengal Plate Test (RBPT), and the diagnostic reagents used were obtained from the Institute of Veterinary Drug Control, China. The procedure with RBPT is briefly described as follows: on a clean plate, 0.03 mL of serum was mixed with 0.03 mL of antigen, and the aggregation results were determined after 4 min.

### 2.3. The Seroprevalence Investigation of Brucella Antibody in Vaccinated Cattle in Lingwu City

The cross-sectional serological survey was conducted in seven townships in Lingwu City from 2021 to 2023. A total of 5435 samples from 622 farms were collected over the course of three years. The serologic test for brucellosis was subsequently conducted.

### 2.4. Investigation of the Positive Rate of Brucella Antibodies in Dairy Herds of Three Brucellosis-Free Dairy Farms

The cows on the three dairy farms were sourced from brucellosis-free regions of Australia and New Zealand, with strict adherence to self-breeding practices. In order to prevent brucellosis, pasture cows were vaccinated with the A19 vaccine, leading to no cases of brucellosis under immune conditions during continuous brucellosis surveillance of cattle for nearly a year. With a robust biosafety system in place, the annual abortion rate across all pastures remained below 5%. The blood and vaginal swab samples were randomly collected from cows that had been immunized with the brucellosis A19 vaccine a minimum of 12 months previously, ensuring a ratio of no less than 10% for random sampling in each ranch. Serologic and etiological testing for Brucella was conducted subsequently. A total of 533 blood and vaginal swab samples were collected, respectively, from three Brucellosis-free dairy farms. PCR was performed using the Mathematical Expectation strategy, in which six samples were mixed into one.

### 2.5. Identification of the DNA Sequence That Distinguishes Brucella A19 from Other Strains

Gene collinearity analysis software Mauve is a Java-based program for gene sequence comparison that incorporates elements from the BLAST program. We utilized the Mauve software to conduct a comparative analysis of the reference sequences of Brucella A19 (NZ_CP030751.1, NZ_CP030752.1) and the classical strain 2308 (NC_007618.1, NC_007624.1) in the NCBI database. The resulting sequence fragments were subsequently verified by the BLAST program on the NCBI website, and then SnapGen software was used to compare the gene sequences across the 68 bp fragments identified from *Brucella abortus* S19 (CP000888.1), A19 (NZ_CP030752.1), 2308 (NC_007624.1), *Brucella melitensis* 16M (AE008918.1), and *Brucella suis* strain 1330 (CP002998.1).

### 2.6. PCR-Based Differential Diagnostic Approach

PCR primers were designed to distinguish A19 from other strains according to the specific sequence differences using Primer5 software. The upstream and downstream primer sequences were 5′-TCGTTCCTTTCGCCCTATTAC-3′ and 5′-TGTTGAAGCCGAGCCAGTC-3′, respectively. The expected amplicon size for strain A19 was 374 bp, while other strains were predicted to yield a fragment of 442 bp.

The specificity of the primers was verified by PCR using genomic DNA from *B. abortus* strain A19, *B. suis* strain S2, *Escherichia coli*, *Salmonella*, *Staphylococcus,* and *Streptococcus* preserved in our laboratory.

The sensitivity of our method was compared with the PCR method based on the BCSP31gene specified in the “Diagnosis of Brucellosis” issued by the National Health Commission of the People’s Republic of China. The measured concentrations of A19 genomic DNA (20 ng/μL) were serially diluted 10-fold, and PCR reactions were detected at concentrations of 2 × 10^−3^, 2 × 10^−4^, 2 × 10^−5^, 2 × 10^−6^, 2 × 10^−7^, 2 × 10^−8^, and 2 × 10^−9^ ng/μL. The sensitivity of the established PCR assay was determined.

The 25-µL PCR system comprised 12.5 µL 2× Taq PCR Mix (Vazyme Biotech Co., Ltd., Nanjing, China), 1 µL each primer, 8 µL double-distilled water, and 2.5 µL template DNA. The PCR amplification was performed using the following parameters: pre-denaturation at 95 °C for 5 min, followed by 35 cycles of denaturation at 95 °C for 60 s, annealing at 58 °C for 30 s, and extension at 72 °C for another 30 s. The amplified products were analyzed on a 1% agarose electrophoresis gel and visualized under ultraviolet light.

### 2.7. Processing of Swab Samples

Total DNA was extracted from vaginal swab samples for PCR amplification using a highly effective commercial DNA extraction kit (Tiangen Biochemical Technology Co., Ltd., Beijing, China) according to the manufacturer’s instructions. The kit utilizes a centrifugal adsorption column and buffer system that can selectively bind DNA, enabling the stable and efficient purification of genomic DNA from swab samples. The amount of DNA obtained from a single swab sample ranged from 0.5 to 3.5 μg.

### 2.8. Investigation of Brucella Field Strains among Cattle in 10 Dairy Farms

A total of 1004 blood and vaginal swab samples were collected, respectively, from cattle that had been immunized with the A19 vaccine for more than 12 months on 10 dairy farms, with a minimum sampling proportion of 10% for random sampling. Serologic and etiological testing for Brucella was conducted subsequently. PCR was performed using the Mathematical Expectation strategy, in which six samples were mixed into one. DNA-positive samples were further characterized using a multiplex PCR method for Brucella species identification; bands of 846 bp, 501 bp, and 399 bp bands were amplified from *Brucella suis*. 618 bp, 501 bp, and 399 bp bands were amplified from *Brucella melitensis*. A 361 bp band was amplified from *Brucella abortus* strain A19. A 399 bp band was amplified from other strains of *Brucella abortus* [38].

### 2.9. PCR Detection of Brucellosis Utilizing the Mathematical Expectation Method

For disease detection, if the result is negative, K samples can be tested once, with an average of 1/K times for each sample. If the result is positive, it means that at least one sample in the group has a positive result. Then the samples were tested one by one, and each sample required (K + 1)/K tests on average. The ME formula is as follows:E(X) = 1/K × (1 − p)^K^ + (1 + 1/K) × (1 − (1 − p)^K^)

Here, E(X) is the total number of tests needed, p is the prevalence, and K is the number of samples mixed in one test. By taking the derivative of the formula, we can calculate the value of K when E(X) reaches the minimum value.

We applied the expected prevalence rate of 3% for brucellosis in livestock within Lingwu City adopted in a previous official sampling strategy to the ME formula and concluded that the number of required detections could be minimized by mixing 6 samples. According to the instructions, 400 μL of each vaginal swab sample were required for DNA extraction. To reduce the impact of mixed samples on sensitivity, 400 μL was used for each vaginal swab sample involved in mixing. The mixed vaginal swab samples were then centrifuged at 8000 rpm for 5 min to concentrate them to a final volume of 400 μL. The total DNA was extracted following instructions, and then PCR was conducted. In the event of a positive result from the pooled sample, each of the six samples was tested individually. A total of 1537 samples in Section 2.4 and Section 2.8 have been tested by PCR using the Mathematical Expectation strategy. The number of reactions during detection was counted.

## 3. Results

### 3.1. Seroprevalence of Brucella Antibody in Vaccinated Cattle

The individual and herd Brucella antibody seropositivity rates in dairy cows immunized for more than 12 months in Lingwu City from 2021 to 2023 are shown in Table 1 and Table 2, respectively. A total of 5435 samples were collected over the course of three years. 1447 samples were seropositive by the RBPT method, with an average prevalence of Brucella antibody seropositivity in dairy cattle of 26.6%. The average herd seropositivity rate for cattle in Lingwu was 38.8%. The individual positive rate and the population positive rate of Brucella antibodies showed an upward trend over three years.

### 3.2. Seroprevalence of Brucella Antibodies in Three Brucellosis-Free Dairy Farms

A total of 533 cows from three dairy farms immunized with the A19 vaccine more than 12 months previously were tested for antibodies (Table 3). Additionally, Brucella molecules were not detected in vaginal swab samples collected from both antibody-negative and antibody-positive cattle. The results showed that a certain proportion of cattle remained persistently antibody positive for more than 12 months after being immunized with the A19 vaccine.

### 3.3. Distinctive Differential Sequences of Brucella abortus Strain A19

The genome comparison software Mauve was used to determine differences in the genome sequences of *Brucella abortus* strain A19 and *Brucella abortus* strain 2308. This comparison revealed a 68 bp sequence deficiency in the genome of the strain A19 between positions 371,765 and 371,766 on Chromosome 2, which corresponds to the sequence between positions 371,759 and 371,826 of the strain 2308 genome. Comparison of the genome sequences of Brucella strains from positions 371,465 to 372,065 using the BLAST program in NCBI showed that only strains A19, LBAB038, S19, and 19BA had 100% similarity compared to other Brucella strains. Thus, our data revealed that only these four Brucella strains lacked the deletion of 68 bp bases identified by Mauve software. The specific deletion sequence of *Brucella abortus* strain 19 is shown in Figure 1.

### 3.4. PCR-Based Identification of Brucella abortus Strain A19

Using the *Brucella abortus* strain 2308 as a template, we used Primer 5 software to design a pair of primers across the 68 bp deleted sequence of strain A19 to establish a PCR method for distinguishing strain A19 from other strains. As shown in Figure 2, the established PCR method generated products of the expected sizes for strains A19, 2308, S2, and 16M. The figure in the manuscript and full-length gels and blots are provided in the Appendix A.

### 3.5. PCR Assay Specificity and Sensitivity

To assess the discriminatory power of the established PCR method among Brucella strain A19, Brucella non-A19 strains, and other non-Brucella bacteria, DNA amplification was performed using these strains as templates. As shown in Figure 3, a 374 bp fragment was amplified from the A19 DNA template, while a 442 bp product was generated from non-A19 Brucella strains, and no amplification signal was detected from the other non-Brucella bacteria. The results showed that the established PCR method had good specificity for the identification of Brucella and other main bovine pathogenic bacteria.

As shown in Figure 4, the established PCR method demonstrated the same sensitivity as the PCR method outlined in the diagnosis of Brucellosis issued by the National Health Commission, achieving a detection limit of 2 × 10^−8^ ng/μL. The results showed that the PCR method for identifying Brucella A19 had good sensitivity.

We next evaluated the established PCR method to identify Brucella field strains in dairy cattle. A total of 1004 samples were collected from cows on 10 dairy farms (Table 4). Brucella field strains were found to be present on 5 of the 10 farms (Figure 5). Furthermore, all 10 Brucella field strains were identified as *Brucella abortus* (Figure 6). *Brucella abortus* was the predominant strain identified in the field during this study.

### 3.6. PCR Detection of Brucellosis Utilizing Mathematical Expectation

The ME method offers the advantage that six samples can be pooled into one for detection purposes. A total of 306 reaction times were performed using the ME method on 1537 samples. This represented a significant decrease in the workload of 80.1% and a greatly enhanced detection efficiency (Table 5).

## 4. Discussion

It is of significant value to conduct investigations on the prevalence rate of Brucella antibodies and the field strains of Brucella in livestock, as this can aid in controlling the spread of brucellosis among livestock, thereby reducing the incidence of human brucellosis and promoting the eradication of this disease [39]. The rate of persistent antibody immunity more than 12 months after Brucella vaccine immunization in NingXia remains unknown. In this study, we aimed to elucidate the serological and etiological status of brucellosis in immunized dairy herds, thereby providing a scientific foundation for epidemiological investigation, diagnosis, control, and eradication of brucellosis in this region as well as other areas where immunization is implemented.

To overcome the challenge of identifying cattle infected with Brucella field strains, we aimed to identify deletion or addition sequences of the Brucella strain A19 by comparing its genome sequence with those of other strains. We identified a 68 bp deletion fragment on chromosome 2 specific to the A19 strain through Mauve software Version 2.4.0, which was consistent with previous reports [40]. Then we subsequently designed upstream and downstream primers to discriminate between A19 and non-A19 strains of Brucella based on the size difference of the target amplified fragment using PCR methodology. The established PCR method can accurately distinguish the major pathogenic bacteria in cattle for specific detection in the laboratory and successfully apply this method to identify cows infected with field Brucella strains in dairy herds. Furthermore, this method can also be employed for identifying 19 and non-19 strains of Brucella in cattle in other vaccination areas. According to the vaccine immunization program implemented from 2021 to 2023, individual Brucella antibody-positive rates among cattle vaccinated in Lingwu City were recorded as 23.4%, 26.7%, and 30.0%, respectively, while herd positivity rates stood at 32.9%, 36.2%, and 55.3%, respectively. The seroprevalence of Brucella antibodies in dairy cows exceeded that reported previously [34], potentially due to the increased utilization of vaccines or the escalated dissemination of brucellosis. Due to the local implementation of the A19 vaccine, traditional serological methods are not suitable for differentiating antibody responses caused by infections with field strains of Brucella [30]. These findings indicate that a certain proportion of cattle immunized with the brucellosis vaccine remain seropositive for more than one year in Lingwu-immunized dairy herds. The achievement of an individual seropositive rate below 0.4% and a population seropositive rate below 7% by 2026 poses significant challenges. The results of the serological survey revealed that an average of 26.6% of cows vaccinated with the immune Brucella A19 tested positive for antibodies after 12 months. We hypothesize that many of these antibodies are persistent and produced by the vaccine itself. It is suspected that the persistent antibody may be attributed to previous immunization (prime and boost) with Brucella vaccines in the Lingwu area and that strain A19 exhibits stronger virulence compared to strain S19 [29]. Therefore, we conducted an additional seroprevalence survey on dairy farms free from brucellosis to ascertain the proportion of antibodies that persist 12 months post-vaccination.

Previous research has shown that this immunization program effectively renders cows negative for brucellosis after 7 months under experimental conditions [38]. However, our investigation of three Brucella-free dairy farms revealed that between 5.4% and 26.5% of cows still tested positive for A19 vaccine-induced antibodies at 12 months post-immunization. Additionally, no evidence of field Brucella strains was found in swab tests conducted on these cattle, and the majority did not exhibit typical symptoms such as abortion associated with brucellosis. Therefore, we infer that these persistent antibodies are durable titers induced by the vaccine, which aligns with previous findings [32,33]. The persistence of vaccine antibody titers is directly related to vaccinating heifer calves after they are eight months of age or older. The mean titers of calves vaccinated at eight months receded to a negative status within approximately 12 months [32]. A proportion of the cattle sampled in this study were vaccinated with a booster vaccine when they were older than 8 months of age. This increased the percentage of cattle that developed persistent antibodies 12 months after immunization. In addition, the presence of these persistent antibodies is likely influenced by factors such as vaccine virulence and immunization schedules. Consequently, serological methods alone cannot accurately diagnose bovine brucellosis in vaccinated areas, nor can arbitrary culling of serologically positive individuals guarantee the elimination of actual cases since a significant number of uninfected Brucella field strain cattle may be unnecessarily destroyed.

A total of 1004 bovine vaginal swabs from vaccinated cattle were collected to detect 10 field strain samples, all of which were further identified as *Brucella abortus*. Since it has been reported that human brucellosis in NingXia is predominantly caused by *Brucella melitensis* [41], our results further suggested that sheep suffering from brucellosis in NingXia are an important source of human brucellosis. Thus, there is a need to strengthen prevention and control measures for brucellosis in sheep to reduce the incidence of human cases. We speculated that the risk of transmission of brucellosis in dairy cattle in Lingwu City was the brucellosis in cattle itself, not other domestic animals. It is particularly noteworthy that Brucella field strains were not detected in the antibody-negative cattle, while those infected with Brucella field strains all tested positive for antibodies, and abortion occurred within 15 days in this study. Previous reports [33] have indicated that cows infected with brucella typically do not transmit the infection to other animals unless calving or aborting, after which they become efficient transmitters via products of parturition. However, their ability to infect other animals diminishes rapidly after delivery, usually within 30 days [33]. Our findings support this perspective and suggest that timely identification of abortion cases can serve as an effective means for the eradication of Brucella-infected cattle through continuous culling. Our results confirmed the positive role of the A19 vaccine in reducing the incidence of brucellosis-induced abortion in cattle by visiting farm ranch workers, but also demonstrated that vaccines cannot be used as the only means of decontamination, as cattle cannot rely on the protective effect of vaccines to clear Brucella field strains. The virulence of the Brucella A19 vaccine, the booster immunization schedule, and the field *Brucella* spp. could potentially be combined, resulting in cattle still testing positive for antibodies 12 months after immunization. Although the vaccine reduced the incidence of miscarriage, cattle that developed persistent antibody titers from the vaccine and cattle that were re-exposed to Brucella but did not become infected and remained positive for the antibody should not be culled. In fact, only cattle infected with field strains of Brucella should be culled. These findings provide valuable data for informing the prevention and control strategy of brucellosis in dairy cattle in areas of dairy herds immunized with the Brucella 19 vaccine.

The previously reported method for mutation screening utilizing the Mathematical Expectation (ME) strategy exhibits both speed and accuracy, making it suitable for large sample sizes with low-frequency mutations [42,43]. Epidemiological investigations usually involve large sample sizes and require time-consuming and labor-intensive testing. In the current study, we were able to reduce the number of tests from 1537 required to analyze all the vaginal swab samples individually using traditional techniques to only 306 by employing the ME strategy, where six samples are mixed into one test, thereby reducing the test workload by 80.1%. Thus, the ME strategy is a time-saving and cost-effective method for molecular detection of brucellosis.

This study also has certain limitations. The RBPT method utilized for the analysis of a large number of serum samples serves as a preliminary screening approach for brucellosis antibodies rather than a diagnostic method. In comparison with the cELISA method, the RBPT method exhibits lower sensitivity [44]. Although the RBPT method has false positives and slightly lower sensitivity, it has the highest specificity in field detection and a high coincidence rate with ELISA [44,45]. Therefore, we speculate that the RBPT method is reliable for large-scale serum detection. In addition, RBPT is more rapid, cost-effective, and suitable for widespread use at the grassroots level compared to cELISA. Nucleic acids had been extracted for PCR analysis from vaginal swab samples collected from cows that had been immunized more than 12 months previously. Due to the fact that field strains of Brucella were not in their excretion period during sampling of infected cattle, these cases may have been missed. Therefore, it is strongly recommended that individual molecular detection be conducted on abortive cattle, and those affected by brucellosis should be eliminated promptly. In this study, we also discovered a significant number of antibody-positive cattle one year after immunization of herds with the Brucella attenuated vaccine; however, not all of these cattle were infected with field strains of brucellosis. There may be three types of persistent antibodies. Firstly, immune antibodies produced by the vaccine can last for more than one year. Secondly, after the immunity wanes, cattle are exposed to a Brucella field strain or vaccine strain, leading to a re-stimulation of antibody production in the cattle without actual infection by the field strain. Third, cattle were infected with Brucella field strains, and antibodies were developed. In the first two cases, although the antibody test was positive, they were not infected with Brucella field strains. According to the results of this investigation, these cattle should account for a large proportion and should not be blindly culled to eradicate brucellosis. Otherwise, it will be resisted by the farmers and cause unnecessary losses [24]. Consequently, investigating the duration of antibodies and influencing factors following immunization with the A19 vaccine should be considered as a future research direction. These vaccine-induced antibodies may interfere with efforts aimed at eradicating brucellosis, and mistaken culling of these cattle would result in substantial economic losses and burdens both for ranches and local governments.

The eradication of brucellosis is a complex and costly endeavor. Many countries, including the United States and Australia, have dedicated decades to achieving success, while many countries have failed due to inadequate policy implementation, insufficient funding, and a lack of awareness among farmers. Vaccination has been implemented in the Northwest of China for several years now. This survey revealed that immunization has significantly reduced the incidence of brucellosis in many farms, confirming the effectiveness of the local vaccine immunization program, although it is crucial to consider factors such as local vaccine immunization programs, vaccine-induced antibody persistence, and epidemic strains of brucella. After a certain period of time, simply relying on serological tests to determine whether cattle are infected with wild strains is not sufficient, and inappropriate criteria for brucellosis diagnosis may result in the unnecessary culling of a significant number of domestic animals, leading to substantial economic losses and impeding the progress of animal husbandry. In summary, we propose that the diagnosis of brucellosis in cattle vaccinated with Brucella A19 should not solely rely on serological methods due to the persistence of vaccine-induced antibodies but rather incorporate PCR methods to detect field Brucella infection.

## 5. Conclusions

This study represents the first comprehensive report on the seroprevalence of brucellosis and the field strains in dairy cattle vaccinated with the Brucella vaccine in Lingwu City. A PCR method was developed to differentiate Brucella strain A19 from other strains, which was successfully employed for detecting field strains of Brucella. The implementation of this ME-based detection strategy can enhance the efficiency of diagnosis, reduce the detection time, and alleviate the workload. Additionally, we propose a diagnostic approach for brucellosis in dairy herds immunized with the A19 vaccine. This study provides valuable data for controlling brucellosis among cows in Lingwu City and offers a reference method for investigating brucellosis epidemiology in livestock populations receiving immune vaccines such as *Brucella abortus* strain 19.

## Figures and Tables

**Figure 1 vetsci-11-00288-f001:**
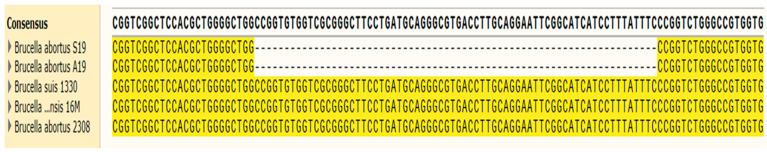
The Brucella 19-specific deletion 68 bp sequences on genome 2.

**Figure 2 vetsci-11-00288-f002:**
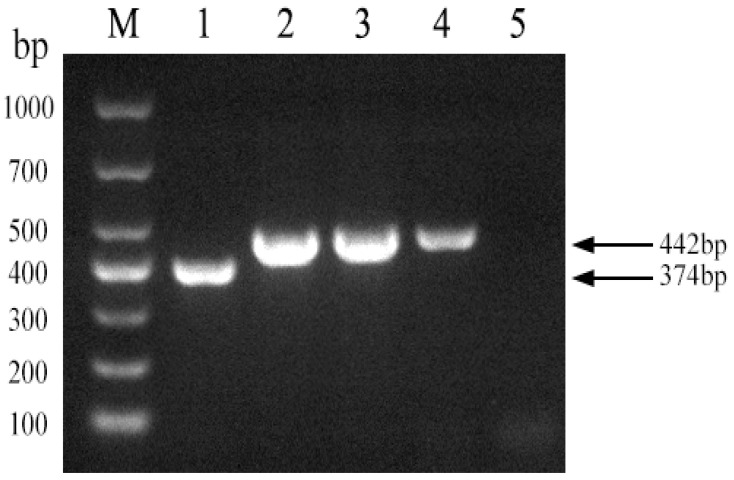
Development of a PCR method to identify Brucella A19 and non-A19 strains. M, DL 1000 DNA Marker; 1, *B. abortus* strain A19; 2, *B. abortus* strain2308; 3, *B. suis* strain S2; 4, *B. melitensis* strain16M; 5, Negative control.

**Figure 3 vetsci-11-00288-f003:**
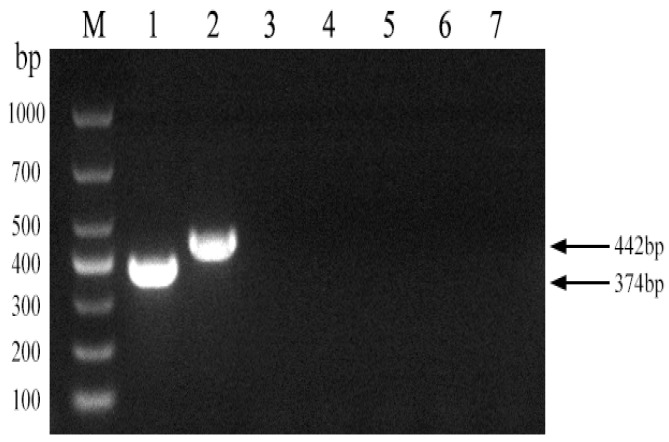
Specificity of the PCR method for identification of Brucella A19 and non-A19 strains. M, DL 1000 DNA Marker: 1, *B. abortus* strain A19; 2, *B. suis* strain S2; 3, *Escherichia coli*; 4, *Salmonella*; 5, *Staphylococcus*; 6, *Streptococcus*; 7, Negative control.

**Figure 4 vetsci-11-00288-f004:**
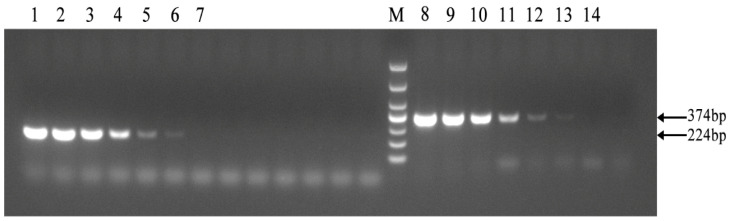
Sensitivity of the PCR method for identification of Brucella A19 and non-A19 strains. M, DL 1000 DNA Marker: The national standard PCR method was used from 1 to 7, and the target fragment size was 224 bp. The PCR method established in this paper was utilized between 8 and 14. The concentrations of Brucella A19 template DNA used in 1–7 and 8–14 were 2 × 10^−3^, 2 × 10^−4^, 2 × 10^−5^, 2 × 10^−6^, 2 × 10^−7^, 2 × 10^−8^, and 2 × 10^−9^ ng/μL, respectively. The detection limits of the two methods were the same.3.6. Examination of Field Strains of Brucella in Cattle from Ten Dairy Farms.

**Figure 5 vetsci-11-00288-f005:**
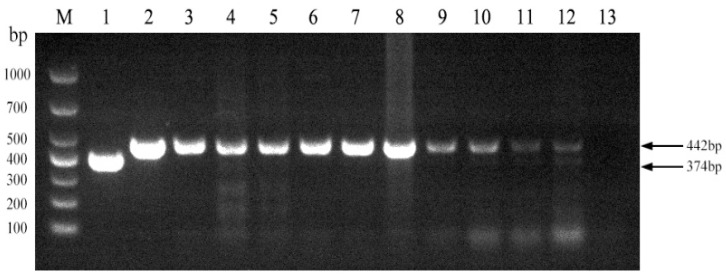
Investigation of Brucella field strains in 10 large-scale dairy farms using the established PCR method. M, DL 1000 DNA Marker; 1, *B. abortus* strain A19; 2, *B. suis* strain S2; 3–12, field strains; 13, Negative control.

**Figure 6 vetsci-11-00288-f006:**
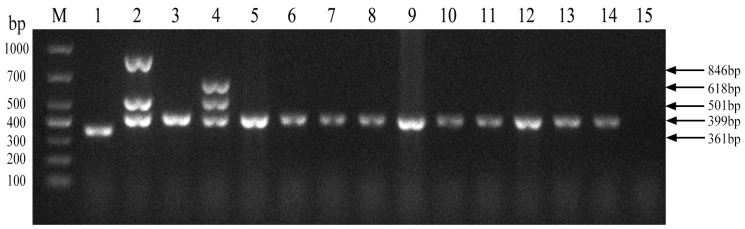
Identification of Brucella field strain species. M, DL 1000 DNA Marker; 1, *B. abortus* strain A19; 2, *B. suis* strain S2; 3, *B. abortus strain 2308*; 4, *B. melitensis* strain 16M; 5–14, field strains; 15, Negative control.

**Table 1 vetsci-11-00288-t001:** Seropositivity rate of dairy cows in Lingwu City from 2021 to 2023.

Year	Number	Number of Positives	Positive Rate
2021	1726	404	23.4%
2022	2146	574	26.7%
2023	1563	469	30.0%

**Table 2 vetsci-11-00288-t002:** Herd seropositivity rates in dairy cows in Lingwu City from 2021 to 2023.

Year	Number	Number of Positives	Positive Rate
2021	234	77	32.9%
2022	265	96	36.2%
2023	123	68	55.3%

**Table 3 vetsci-11-00288-t003:** Seroprevalence of Brucella in three brucellosis-free dairy farms.

Farm	Number	Serological Testing	
		Positive Number	Positive Rate
A	253	67	26.5%
B	150	27	18.0%
C	130	7	5.4%

**Table 4 vetsci-11-00288-t004:** Results of detection of the Brucella field strain in 10 dairy farms.

Farm	Number	Serological Testing		PCR Testing	
		Positive Number	Positive Rate	Positive Number	Positive Rate
1	185	32	17.3%	0	0.0%
2	153	23	15.0%	0	0.0%
3	143	15	10.5%	0	0.0%
4	106	16	15.1%	0	0.0%
5	69	23	33.3%	2	2.9%
6	97	40	41.2%	2	2.1%
7	75	27	36.0%	1	1.3%
8	40	12	30.0%	0	0.0%
9	56	23	41.1%	2	3.6%
10	80	31	38.8%	3	3.8%

**Table 5 vetsci-11-00288-t005:** Statistics of reaction times based on the ME method.

Breeds	Data
Sizes	1537
Assumed prevalence	3.0%
Number of individuals in one reaction time (NR1)	1
Reaction times (RT1)	1537
Number of individuals in one mixed group (NG6)	6
Reaction times (RT6)	306
Reduction rate (RR)	80.1%

## Data Availability

The raw data used and/or analyzed during the current study are available from the corresponding author upon reasonable request.

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
