# Peer review of "Investigation of the Seroprevalence of Brucella Antibodies and Characterization of Field Strains in Immunized Dairy Cows by B. abortus A19"

_vetsci, 2024, doi:10.3390/vetsci11070288_

Round 1
Reviewer 1 Report
Comments and Suggestions for Authors
The manuscript is on an important topic for both cattle and human health. The aim of the research is not clear, however. Is it,
- establishment of PCR for verifying the presence of the pathogen in Brucella vaccinated animals, for better livestock management?
- efficacy (or inefficacy) of current Brucella vaccines?
- or both?
- or...?
A clear objective statement would make the manuscript much more relevant.
In addition, the Results are not well presented. Upon reading the manuscript, one feels they are shown at random.
-------
Few examples of necessary editing:
lines 54-55: '...brucellosis is serious public health threat...'
'...brucellosis is a serious public health threat...'
lines 72-73: '...by immunity accumulates naturally or through vaccination...'
'...by immunity accumulated naturally or through vaccination...'
lines 83-87 'However, most vaccines are smooth strains, as most serologic tests are primarily based on detection of antibodies against the O-side chain of Brucella, discriminating between vaccine-induced 85 antibodies and those generated by field strains infection poses a challenge[29, 30]. Therefore, bacterial culture and nucleic acid amplification are necessary for the diagnosis of brucellosis in animals.'
Suggestion: 'However, most vaccines are made with Brucella smooth strains, as most serologic tests are primarily based on detection of antibodies against the O-side chain of Brucella. Discriminating between vaccine-induced antibodies and those generated by newly field strains infection, poses a challenge[29, 30], becoming necessary direct detection techniques such as bacterial culture and/or nucleic acid amplification for the diagnosis of brucellosis in animals.'
line 188: 'A total number of 1004 samples blood and vaginal swab samples were collected...'
Suggestion: 'A total of 1004 blood and vaginal swab samples were collected...'
line 189: '...that had immunized...'
Must be: '...that had been immunized...'
In ' Results':
line 214: '3.1. The Seroprevalence Investigation of Brucella Antibody in Vaccinated Cattle in Lingwu City'
'3.1. Seroprevalence of Brucella Antibodies in Vaccinated Cattle'
lines 221-222: 'Table 1. Investigation on individual Brucella antibody seropositivity rate in dairy cows in Lingwu City from 2021 to 2023.'
'Table 1. Seropositivity rate of dairy cows in Lingwu City from 2021 to 2023.'
lines 223-224: 'Table 2. Investigation on herd Brucella antibody seropositivity rates in dairy cows in Lingwu City 223 from 2021 to 2023.'
'Table 2. Herd seropositivity rates in dairy cows in Lingwu City from 2021 to 2023.'
line 230: 'Table 3. Results of seroprevalence of Brucella antibodies in three brucellosis-free dairy farms.'
'Table 3. Seroprevalence of Brucella in three brucellosis-free dairy farms.'
lines 270-271: '...all 10 Brucella field strains were identified (Figure 4), further identification as Brucella abortus (Figure 5).'
'...all 10 Brucella field strains were identified as Brucella abortus (Figure 5).'
lines 272-273: 'In particular, the cows that tested positive for the Brucella field strains were all found to be antibody-positive within 15 days of abortion.'
Note: Non-sense sentence, or out of context.
lines 283-... '3.7. Molecular Detection of Brucellosis Utilizing Mathematical Expectation'
Note: Is this result from serological or PCR testing? In addition, no mention of this result in Materials and Methods.
Some statements in 'Discussion' may be more appropriate in 'Materials and Methods', or in 'Results', e.g., "For this purpose, we used the gene collinearity analysis software Mauve, a JAVA-based program for gene sequence comparison that incorporates elements from the BLAST program." (lines 301-303).
And many other issues.
The manuscript may be accepted provided it is thoroughly revised and edited.
Comments on the Quality of English LanguageThe quality of the English language is fine. Need to check all the small mistakes throughout the manuscript, as shown in the Comments and Suggestions.
Author Response
|
3. Point-by-point response to Comments and Suggestions for Authors |
||||||||
|
Comments 1: The manuscript is on an important topic for both cattle and human health. The aim of the research is not clear, however. Is it, - establishment of PCR for verifying the presence of the pathogen in Brucella vaccinated animals, for better livestock management? - efficacy (or inefficacy) of current Brucella vaccines? - or both? - or...? A clear objective statement would make the manuscript much more relevant.
|
||||||||
|
Response 1: Thanks for reviewer’s kindly comment. This research aims to provide data to support the development of strategies for preventing, controlling, and eradicating brucellosis by investigating the real positive rate of Brucella antibodies and field strains of Brucella spp. after immunization over 12 months in dairy cows. We have added” This research aims to provide data to support the development of strategies for preventing, controlling, and eradicating brucellosis” research objective to the 35-line abstract to help readers better understand the research goal. The established PCR method is a molecular investigation technique used to distinguish the Brucella A19 strain from other strains. It helps determine whether the source of infection in cattle infected with brucellosis is from the vaccine strain or from other field strains. We have added” The established PCR method for distinguishing between Brucella abortus strain 19 and non-19 strains can be specifically utilized for detecting natural Brucella infection in immunized cattle” research objective to the 51-line abstract to help readers better understand the goal of the established PCR. The established PCR method can accurately diagnose brucellosis in cattle, eliminate naturally infected cattle, and thus better manage cattle herds. Our interviews with farmers confirmed that the A19 vaccine significantly reduced herd morbidity, as noted in line 415 of the article. However, Brucella-infected cattle were still detected among the immunized cattle, indicating that the vaccine does not provide 100% protection and can only reduce the incidence of brucellosis abortion. Therefore, it is necessary to use the PCR method established in this study to specifically diagnose brucellosis in cattle and cull naturally infected cattle.
|
||||||||
|
Comments 2: In addition, the Results are not well presented. Upon reading the manuscript, one feels they are shown at random. |
||||||||
|
Response 2: We thank the reviewers for their relevant suggestions. We have made revisions to the manuscript. In line 41 of the abstract, we included the results of the PCR assay based on the ME method, which had not been presented before. Add conclusions about “There was a certain proportion of cows with a persistent antibody titer, but there was no evidence that all of these cattle were naturally infected with Brucella” and “Brucella abortus was the predominant strain identified in the field during this study. Detection based on the Mathematical Expectation strategy can significantly enhance detection efficiency” in line 49 of the abstract to help readers better understand the research conclusions. Results related to “The individual positive rate and the population positive rate of Brucella antibody showed an upward trend over three years” are included in line 251 of Section 3.1 Results related to “The results showed that a certain proportion of cattle remained persistently antibody positive for more than 12 months after being immunized with the A19 vaccine.” are included in line 261 of Section 3.2. In line 199 of Section 2.6, we included a sensitivity description test for the established PCR method. In Section 3.5, the detection limit of the established PCR method was 2×10-8 ng/μL, which was equivalent to that of the national standard method, demonstrating that the established PCR method exhibited good sensitivity. Results related to “Brucella abortus was the predominant strain identified in the field during this study” are included in line 316 of Section 3.6 The revised content may help readers better understand the results and conclusions of this study.
|
||||||||
|
4. Response to Comments on the Quality of English Language |
||||||||
|
Point 1: The quality of the English language is fine. Need to check all the small mistakes throughout the manuscript, as shown in the Comments and Suggestions. |
||||||||
|
Response 1: We thank the reviewers for recognizing the quality of English language in the article. We accepted the reviewer's comments on the English editing of the article and made changes. We also made some edits in English. All modifications are tracked and highlighted in the article. |
||||||||
|
5. Additional clarifications |
||||||||
|
Please allow me to outline this study so that readers can better understand its content: |
â‘´ Investigation of the positive rate of Brucella antibody: From 2021-2023, serological survey was conducted in cows immunized with Brucella A19 for more than 12 months in Lingwu city. All the sampled cows were taken as individuals and each dairy farm or farmer was taken as a herd. The individual positive rate and herd positive rate of Brucella antibody were calculated during the three years.
⑵ Demonstration of persistent antibody of brucellosis vaccine: Serological survey was carried out in three brucellosis- free dairy farms, and molecular detection of the pathogen was performed by using the established PCR method. It was concluded that the immunization program in this area could lead to the presence of a certain proportion of persistent antibody in cattle. We propose that the detection of antibodies in cattle vaccinated with the Brucella A19 vaccine more than 12 months previously should not be solely relied upon as a diagnostic basis for brucellosis.
â‘¶ Investigation of Brucella field strains in large-scale dairy farms in Lingwu city: First, the specific recognition sequence of Brucella A19 was found, and the primers were designed to establish a PCR method. After verifying the feasibility of the PCR method, the field strains of Brucella were investigated in 10 randomly selected pastures with different geographical distribution. Finally, the characteristics of Brucella strains in this area were obtained.
â‘· ME method: According to the formula based on the prevalence of 3%, when 6 samples were mixed into 1, the total number of tests could be minimized. Following this strategy successfully reduced the workload throughout the pathogen molecular PCR detection process.
⑸ The positive rate of Brucella antibody and the characteristics of field strains provide the basis for the prevention and control of brucellosis in Lingwu city. This study provides a successful method for identifying the pathogenic molecules of Brucella 19; It is clear that cattle immunized more than 12 months with antibody positive should not be all identified as brucellosis livestock, so as to avoid mistakenly killing a large number of domestic animals due to vaccine persistent antibody. The use of ME method can significantly reduce the workload.

Reviewer 2 Report
Comments and Suggestions for Authors
The target of the manuscript is to evaluate persistence of antibody 12 months after vaccination with Brucella A19 strain. It is not clear that the design is the best suited for the purpose. Persistence over time could be evaluated testing same animals over a period of time (longitudinal study to follow the kinetics); ideally, these can be tested with several diagnostic methods in parallel at the different time points.
The study is based on Rose Bengal plate test – Why complement fixation test was not used?
Route and age of vaccination can have an impact on the duration of serology. Text describes (line 126) that sampling was performed in cows older than 1.5 years (18 months). According to discussion (line 313) animals are vaccinated between 3 and 8 months of age, followed by a low-dose booster at 3 months later; this gives a range of 6 (3+3) to 11 (8+3) months to complete vaccination. To check these vaccinated animals 12 months later – this would require and age of 18 to 23 months – this means that animals in the study do not fulfill the criteria (?). Can these differences in age be related to differences in antibody prevalence? I think that the only way to solve this is to show results analyzed regarding prevalence and months after vaccination.
Regarding sampling for PCR – the numbers are not consistent, and the criteria for selection of animals are not clear (percentage of abortions in the herd?). Pooling samples (line 208, line 376) requires to demonstrate that the analytical sensitivity is not affected (i.e. by pooling a weak positive with the number of negative samples) and an extensive validation. Note that the approach is different than the one described at 369-371 – which involves mixing equal amounts of quality DNA
Were PCR results compared with other differential PCRs such as AMOS and Bruce-ladder?
Line 56: References 7 and 8, regarding estimates for human cases are 2006 and 2010 – if available, use updated references
Line 63: “In recent years, 12 new species of Brucella have been identified[12]” This is not clear: 12 plus the classical ones, or 12 in total ? Formally, only 12 species are cited in the OMSA manual.
Line 65: “B. melitensis, B. abortus, B. suis, and B. canis are the main causes of the disease in humans,…” All species equally relevant?
Line 70: The sentence (“Individual immunity refers to a physiological function of the body's immune system to maintain the body's health” can be removed.
Line 124: “… and farmers…” this is not clear- were also the farmers (owners, staff) tested for serology?
Line 128: “… each dairy farm or farmer was taken as a herd ….”, please, explain.
Line 142: The commercial Rose Bengal Plate Test (RBPT) for Brucella should briefly described; mention if this is according the OMSA Manual.
Was the PCR-based differential diagnostic designed for this manuscript or has been already described/used elsewhere?
Also, original source of A19 and manufacturer should be mention.
Line 174: “other strains…” Which strains? Does it refer to in-silico estimation? Was it evaluated on other species (B. melitensis)
Line 178: 1 μl each primer - add concentration
Line 185: the name of the commercial DNA extraction kit must be included. Describe the reason to mention “highly effective”
Line 192: The multiplex PCR method for Brucella species identification should be briefly described. Ref [37] not found in internet.
Line 96: Route and age of vaccination can have an impact on the duration; this may be commented here? Also, it can be useful to mention that there is not a test method available of immune status in animals or populations post-vaccination (see table 1, chapter 3.1.4 OMSA Manual).
Line 208: As commented before, pooling samples requires a validation. Was pooling performed before or after DNA extraction?
Line 211: “… 1537 collected samples were tested” the number is not consistent with line 188.
Line 261 (also lines 307-309): “These results indicated the high specificity of this PCR method …. “ This statement would require extensive testing with species/serovars.
Line 273: “… within 15 days of abortion” this criteria for the sampling has not been explained before.
Line 361: “Our findings support this perspective …” Please, explain.
Discussion section would revision. Are these results extrapolated to other vaccines, such as S19?
Minor issues
Line 11, 409: to achieving … to achieve
Line 117: from in immunized dairy … from vaccinated dairy
Line 188: samples blood … blood samples
Line 344: in immune areas … in vaccinated areas
Line 397: the cattle vaccine failed to be immunized – check sentence
Line 411: (Zhang, 2018) … [9]
Comments on the Quality of English LanguageNo specific comments.
Author Response
|
3. Point-by-point response to Comments and Suggestions for Authors |
||||||||||||||||||||
|
Comments 1: This paper describes a study to determine the brucellosis seroprevalence of cattle that are vaccinated against brucellosis using B. abortus A19 strain in Lingwu city of NingXia Province of China. The subject animals were older than 1.5 years and have received the vaccine at least over a year ago. Three main studies have been described: â‘ Seroprevalence in dairy cattle of the Lingwu City: samples from ~5500 animals from 622 farms collected over 2.5 years were tested. Observations: On average 26% animals and 38% farms were seropositive â‘¡ Seroprevalence in Brucella free animals: ~500 animals from 3 presumably Brucella free farms were tested (serum for antibodies and vaginal swabs for Brucella DNA). Observations: Individual animal seropositivity ranged 5-26% while 100% (3/3) was seropositive at the farm level. No PCR positive vaginal swabs. â‘¢ Seroprevalence and Brucella pathogen prevalence on selected farms: ~1000 animals from 10 farms were tested (serum for antibodies and vaginal swabs for Brucella DNA.) Individual animal level seropositivity was 10-41% between farms and 100% (10/10) farms were seropositive. Vaginal swabs of 50% farms were B. abortus DNA positive by PCR |
||||||||||||||||||||
|
Response 1: Thanks for the correct summary of the contents of this study. These survey data also provide fundamental information for local government departments to develop prevention, control, and eradication measures for bovine brucellosis, which is main of the objectives of this study. |
||||||||||||||||||||
|
Comments 2: All 13 farms that were selected for detailed study (serology and PCR) were seropositive. However, when the total samples were considered, only 32-55% of the farms were seropositive over the course of the study. Assuming the farms were selected randomly for all studies, it may be of interest to see any explanation on this discrepancy. |
||||||||||||||||||||
|
Response 2: In line 149 of Section 2.1, we increased the data that pathogen investigation from March to May 2023 and randomly selected 10 dairy farms situated in all 7 districts of the city. Ten ranches had more than 100 animals, excluding farmers with small numbers of dairy cattle. The herd positive rate of Brucella in 2023 was 55.3%; therefore, the herd positive rate of Brucella antibody in 10 pastures was deemed reasonable. A certain proportion of cattle with positive antibodies were also found in the three pastures assumed to be free of brucellosis, indicating that not all of the cattle with positive antibodies were naturally infected with Brucella. This finding poses a challenge for how to cull cattle that test positive for the antibody.
|
||||||||||||||||||||

Reviewer 3 Report
Comments and Suggestions for Authors
This paper describes a study to determine the brucellosis seroprevalence of cattle that are vaccinated against brucellosis using B. abortus A19 strain in Lingwu city of NingXia Province of China. The subject animals were older than 1.5 years and have received the vaccine at least over a year ago.
Three main studies have been describe,
1. Seroprevalence in dairy cattle of the Lingwu City: samples from ~5500 animals from 622 farms collected over 2.5 years were tested. Observations: On average 26% animals and 38% farms were seropositive
2. Seroprevalence in Brucella free animals: ~500 animals from 3 presumably Brucella free farms were tested (serum for antibodies and vaginal swabs for Brucella DNA). Observations: Individual animal seropositivity ranged 5-26% while 100% (3/3) was seropositive at the farm level. No PCR positive vaginal swabs.
3. Seroprevalence and Brucella pathogen prevalence on selected farms: ~1000 animals from 10 farms were tested (serum for antibodies and vaginal swabs for Brucella DNA.)
Individual animal level seropositivity was 10-41% between farms and 100% (10/10) farms were seropositive. Vaginal swabs of 50% farms were B. abortus DNA positive by PCR
This study provide and insight into Lingwu cattle population in terms of their seroprevalence and Brucella pathogen prevalence of ~200,000 cattle population. How representative the data may depend on how representative was the sampling (in terms of the geographical location and distribution of the cattle population, etc.), which is not very clear in this paper. Nonetheless, given this is the first of such report for this region, it provides an important insight. This paper provide supporting evidence to previously shown phenomena that A19 vaccinated animals are seropositive even after 12 months post vaccination and vaccinated animals could still harbor the Brucella organism. Knowledge on the level of seroprevalence and pathogen prevalence within the specific region will help better planning of the eradication programs.
All 13 farms that were selected for detailed study (serology and PCR) were seropositive. However, when the total samples were considered, only 32-55% of the farms were seropositive over the course of the study. Assuming the farms were selected randomly for all studies, it may be of interest to see any explanation on this discrepancy.
Detection of Brucella DNA in vaginal swab by PCR, although specific, may not be sensitive enough to identify every infected animal. Some infected animals may not shed the organism in the vaginal secretions at the time of sampling. Therefore, the observed prevalence could be an underestimation of the true prevalence. Even with that limitation, five out of ten farms had infected animals. Such high level of prevalence may have played a significant part along with the vaccination in the high seroprevalence in this population.
The RBPT was used on pooled serum samples for the initial screening. Mathematic modeling shows the logical number of sample in each pool but it does not consider the effect of pooling on the assay performance (i.e. impact on RBPT sensitivity). While current approach does not make the results invalid, showing the effect of sample pooling on the assay performance may help better interpretation of the serological results.
Line 307: the claim that the newly developed PCR method has been “validated” with “excellent specificity” is misleading. According to the data presented, the authors have demonstrated the assay functionality only with a few field isolates. As the authors claim, if this assay were to use in field studies and clinical trials, a proper validation exercise is very important.
On the same topic, the reference provided for the multiplex PCR used in lines 191-196 is not traceable. Please provide a traceable reference or describe it in detail. Furthermore, newly develop PCR in section 2.6 does not add up to 25ul. Please revise.
Figure 5 is missing B. abortus non-vaccine strain control.
Reference 2 is just a commentary on an error on the reference 1 - Although I am not against citing this paper, authors could easily find a better reference to support the statement of “Brucellosis is caused by bacteria of the genus Brucella” J
Round 2
Reviewer 1 Report
Comments and Suggestions for Authors
The revised and "cleaned" manuscript version still has many grammar and typographical mistakes, such as:
- lines 87-91
"The strains A19 and S19 are predominantly utilized for Brucella vaccination in cattle[28], both originating from Brucella abortus strain19 isolated in the United States in 1923, exhibited a 99.9% homology, with S19 "lacking the 702-bp deletion" in the erythritol gene that was present in A19[29]."
Suggestion: The strains A19 and S19 are predominantly utilized for Brucella vaccination in cattle[28]. Both originated from the Brucella abortus strain 19, isolated in the United States in 1923. Their genomes exhibit a 99.9% homology, with S19 having a 702bp deletion in the erythritol gene, which gene is complete in A19[29].
line 98
mplification0020
line 127
...prevalence of brucellosis and characterize field Brucella spp. "from" in immunized dairy...
line 151
...verify the "persistent" antibody. -> ...verify the presence of anti-Brucella antibodies?
line 162
...testing with "a" commercial Rose Bengal Plate... -> ...testing with the commercial Rose Bengal Plate...
line 163-164
The procedure "of" RBPT "was" briefly described as follows: on "the" clean plate,...
-> The procedure "with" RBPT "is" briefly described as follows: on "a" clean plate,...
THESE ARE ONLY FEW EXAMPLES. MUST REVIEW ALL THE MANUSCRIPT FOR THESE GRAMMAR AND TYPOGRAPHICAL MISTAKES.
Comments on the Quality of English LanguageFew grammatical mistakes, otherwise, the English Language is fine.
Reviewer 2 Report
Comments and Suggestions for Authors
Age of vaccination can have an impact on the duration of serology. The manuscript describes that sampling was performed in cows older than 1.5 years (=18 months). Line 138: “… first immunized with the Brucella A19 vaccine between 3 to 8 months of age, followed by a low-dose booster at 3 months later.” This implies that cattle are 6 (3+3 later) to 11 (8+3 later). Accordingly, 12 months later are 6 + 12 =18 months, only this group of animals would fulfill the criteria. However, the rest of animals do not, e.g. animals vaccinated when 11 months old, if tested when 1.5 years (18 months), this means that is seven months after second dose of vaccine. Can these differences in age be related to differences in antibody prevalence? The way to solve this is to show results analyzed regarding prevalence and months after vaccination.
Route of vaccination may also have an impact. What is the route of vaccination? The same in all animals?
The target of the manuscript is to evaluate persistence of antibody 12 months after vaccination with Brucella A19 strain. It is not clear that the design is the best suited for the purpose. Persistence over time could be evaluated testing same animals over a period of time (longitudinal study to follow the kinetics); ideally, these can be tested with several diagnostic tests in parallel at the different time points.
The study is based on Rose Bengal plate test. The commercial Rose Bengal Plate Test (RBPT) for Brucella should briefly described; mention if this is according the OMSA Manual. It could be useful to test at least a subpanel of those samples with complement fixation test or ELISA.
Regarding sampling for PCR, pooling samples requires to demonstrate that the diagnostic sensitivity is not affected (i.e. by pooling a weak positive - DNA from swab - with the number of negative samples) and the adequate validation.
Results of this study should be compared to refs 31 and 32, or other available data.
Other issues
Title: insert the vaccine name B. abortus A19
The original source of A19 and manufacturer should be mention in the manuscript.
line 66: livestock industry animals … livestock industry
line 68: if available, add updated references
line 74: 12 new species of Brucella have been identified[12]. ]” This is not clear: 12 plus the classical ones, or 12 in total ? Formally, only 12 species are cited in the OMSA manual.
line 81: delete this sentence - Individual immunity refers to a the physiological function of the body's immune sys-81 tem to maintain the body's health.
line 107: immune antibodies … antibodies
line 136: almost whole-dairy cattle – these words are not clear
line 137: farms and farmers, line 142: each dairy farm or farmer was taken as a herd.
I assume this is problem of meaning or translation but these words are not clear: farmer is the owner of the farm. Were people also tested?
line 140: Object of sampling of cows older than 1.5 years … check grammar
line 150: pastures – farms ? The criteria for selection of animals are not clear (percentage of abortions in the herd?)
line 152: check grammar
line 212: name of product should be added; briefly describe the reason to mention “highly effective”
line 221: multiple … multiplex
line 221: multiple PCR method for Brucella species identification should be briefly described and a reference should be added (Ref 37 is not available in internet)
line 259: Brucella molecules … Brucella strain 2308 of as a template DNA
line 279: strain 2308 of as a template … strain 2308 as a template
line 289, and Fig.4: sensitivety … sensitivity
line 295: high specificity – this evaluation includes only a limited panel of bacteria, this statement is not correct.
line 313: check grammar
line 316: this sentence is redundant (line 314)
line 332: check grammar
line 396: immune areas … vaccinated areas
line 424: check grammar
line 456: cattle vaccine failed to be immunized – check sentence
Figures: There are two images for Figure 6
Comments on the Quality of English LanguageSuggestions has been included in previous comments - other issues (such as check grammar)
